# Genetic Diversity of Phenotypic and Biochemical Traits in VIR Radish (*Raphanus sativus* L.) Germplasm Collection

**DOI:** 10.3390/plants10091799

**Published:** 2021-08-29

**Authors:** Anastasia B. Kurina, Dmitry L. Kornyukhin, Alla E. Solovyeva, Anna M. Artemyeva

**Affiliations:** N. I. Vavilov All-Russian Institute of Plant Genetic Resources (VIR), 190031 St. Petersburg, Russia; dkor4@yandex.ru (D.L.K.); alsol64@yandex.ru (A.E.S.); akme11@yandex.ru (A.M.A.)

**Keywords:** *R. sativus*, collection, trait, variability

## Abstract

Small radish and radish are economically important root crops that represent an integral part of a healthy human diet. The world collection of *Raphanus* L. root crops, maintained in the VIR genebank, includes 2810 accessions from 75 countries around the world, of which 2800 (1600 small radish, 1200 radish) belong to *R. sativus* species, three to *R. raphanistrum*, three to *R. landra*, and four to *R. caudatus*. It is necessary to systematically investigate the historical and modern gene pool of root-bearing plants of *R. sativus* and provide new material for breeding. The material for our research was a set of small radish and radish accessions of various ecological groups and different geographical origin, fully covering the diversity of the species. The small radish subset included 149 accessions from 37 countries, belonging to 13 types of seven varieties of European and Chinese subspecies. The radish subset included 129 accessions from 21 countries, belonging to 18 types of 11 varieties of European, Chinese, and Japanese subspecies. As a result of the evaluation of *R. sativus* accessions according to phenological, morphological, and biochemical analyses, a wide variation of these characteristics was revealed, which is due to the large genetic diversity of small radish and radish of various ecological and geographical origins. The investigation of the degree of variation regarding phenotypic and biochemical traits revealed adaptive stable and highly variable characteristics of *R. sativus* accessions. Such insights are crucial for the establishment and further use of trait collections. Trait collections facilitate germplasm use and contribute significantly to the preservation of genetic diversity of the gene pool.

## 1. Introduction

Small radish and radish belong to the botanical species *Raphanus sativus* L. of the family Brassicaceae Burnett (Cruciferae Juss.) [1,2,3]. *R. sativus* is not found in the wild. The origin of the species is unknown. According to most opinions, *R. sativus* derives from wild radish *R. raphanistrum* L. or was obtained by hybridization between *R. maritimus* Smith. and *R. landra* Mor. ex DC. [4,5]. Sinskaya (1928) [1] suggested that direct wild ancestors of cultivated forms of this genus became extinct. According to Vavilov, there are four centers of origin of *R. sativus*: South-West Asia, East Asia, the Mediterranean, and South Asian Tropics [6,7]. *R. sativus* is characterized by the presence of small radish, a dwarf mutant form. Artificial selection was carried out on the basis of dwarfism of plants during vegetative period of ontogenesis. In the reproductive period, small radish is indistinguishable from regular cultivars that do not differ in structure and size from radish.

Multiple genetic processes, such as recombination, mutations at the chromosomal level, inactive gene expression, and altered allele frequencies of phenotypical traits, have been proven to have a major effect on the phylogenetic distance of radish and small radish. Natural and artificial selection in various ecological and geographical conditions has caused large intraspecific diversity in the growth types of *R. sativus*, which is quite remarkable at the diploid level of development [8].

Cultivated radish accessions exhibit diverse morphological and physiological characteristics and are taxonomically categorized as distinct varieties [9,10,11,12].

Small radish and radish are economically important root crops that represent an integral part of a healthy human diet. Roots, young pods, seedlings, and, less often, leaves are used for consumption. Root vegetables contain a variety of phytochemicals, such as phenols, vitamins, and glucosinolates, which are potentially beneficial for human health [13,14]. Due to the presence of these components, small radish and radish are used as an additional therapeutic agent in the diet of patients with diabetes, several types of cancer, cardiovascular diseases, as well as liver and respiratory diseases [15,16,17,18,19].

Radish is grown all over the world, especially in China, Japan, and Korea. Japan annually produces 3.7 million tons of radish daikon, and also imports 0.9 million tons from other countries. In China, radish is cultivated on 1.2 million hectares, which occupies 6% of the total sown area for vegetable crops. Production in 2016 amounted to 44.6 million tons, which corresponded to 47% of the world radish harvest [20,21]. In America, Africa, and Australia, radish is grown all year round, small radish is grown mainly in the spring, and both crops are eaten fresh. Winter radish is mainly grown in France, the Czech Republic, Germany, and the Netherlands. In these countries, an extensive assortment of winter and summer radishes and small radishes has been created. Many European cultivars are widely distributed in America, Africa, and Australia. According to the FAO data (2019), small radish is grown in the world on an area of 70,773 hectares [21]. The main production area is located in Central America (more than 18 thousand hectares) [22,23].

In the Russian Federation, most of the production of radish and small radish is concentrated on the fields, mainly in farms and private kitchen-gardens. Large vegetable-growing greenhouse complexes grow small radish as an intermediate crop or use it for intercropping. At the same time, the commercial production of small radish is insufficient, and the niche during the winter-spring period is covered by imports from European countries [24]. Small radish is grown on an area of more than 4000 hectares, of which about 400 hectares are in greenhouses. The main zones of small radish production in Russia are: South (Krasnodar, Stavropol, and Rostov Regions) and the Central Russia region [25].

Radish breeding has been practiced for centuries, by means of mass or pedigree selection. The most important goal of root vegetable crop breeding programs is to create F1 hybrids and cultivars for year-round consumption, with consistently high yields, high consumer qualities, resistance to biotic and abiotic stresses, and improved biochemical composition, adapted for industrial technologies and suitable for processing, on the basis of traditional and advanced methods of breeding [26]. Therefore, a permanent and extensive study of the historical and modern gene pool of *R. sativus* root crops and establishment of new source material for breeding is necessary [26]. It should also be noted that the methods of evaluating the breeding material and identifying the sources of economically valuable traits for the *R. sativus* root crops breeding are not sufficiently developed. The physiological responses of different variety type accessions to growing conditions, biochemical features, limits of variability of traits, value for breeding, ecological plasticity, adaptive capabilities, and resistance to abiotic and biotic stresses are not sufficiently studied. Modern breeding has significantly narrowed the historical diversity of *R. sativus*. To expand the genetic basis of modern varieties, it is necessary to involve the most diverse genetic material in breeding.

The material for such studies is provided by the world collection of the *R. sativus* root crops of the Russian Federation, maintained in the VIR genebank. The world collection of *Raphanus* L. root crops, maintained in VIR genebank, includes 2810 accessions from 75 countries of the world, of which 2800 (1600 small radish, 1200 radish) belong to *R. sativus* species, three to *R. raphanistrum*, three to *R. landra*, and four to *R. caudatus*. Every year, the VIR collection is replenished with numerous accessions of the modern breeding, primarily from China, Japan, Netherlands, Russia, as well as accessions collected during collection missions primarily in Central Asia and Transcaucasia. New accessions have different status-local populations, breeding varieties, genetic lines, and F1 hybrids.

According to the European Search Catalogue for Plant Genetic Resources (EURISCO, https://eurisco.ipk-gatersleben.de/apex/f?p=103:1 accessed on: 5 February 2021) for 2021, the collection of *R. sativus* in European genebanks counts 3513 accessions. The largest collections are allocated in the genebank of Great Britain (WARGRU Warwick) with 1350 accessions, Germany (IPK–Gatersleben) with 661 accessions, and the Netherlands (CGN–Wageningen) with 308 accessions. The genebank of Japan (NARO Genebank) has a collection of 441 accessions, the United States genebank (U.S. National Plant Germplasm System) 687, while India maintains near 300 accessions. The largest collection of *Raphanus* is stored in the National Genebank of Vegetable Crops at the Institute of Vegetable and Flower Crops in China, with more than 2600 accessions, which are represented by local forms, populations, and F1 hybrids.

The purpose of our study was to investigate the phenotypical diversity of *R. sativus* accessions from the VIR collection according to a set of traits and to identify the degree of their variability and connection with their botanical, biological, and agronomical characteristics, as well as to identify sources of valuable traits.

## 2. Results

### 2.1. Variability of Phenotypic Traits

The extent of variability present in the radish types was measured in terms of range, mean, and coefficient of variation (CV). All investigated types differed significantly with respect to all studied characteristics.

Phenological observations revealed a difference in the time of onset of technical ripeness in the different accessions of small radish and radish.

In our studies, the total variability range for the duration of vegetative phase in small radish was 20.5–37.5 days (CV = 14.7%), and in radish 30.0–87.0 (CV = 28.1%) (Figure 1). The small radish collection was divided into nine groups with a two-day interval, and the radish collection into eight groups with a seven-day interval. On the basis of the results, it was decided to combine the last five groups of small radishes into two resulting groups. Furthermore, small radish was divided into five resulting groups. These are defined as ultra-ripe (20–22 days), early-ripe (22–24 days), medium-ripe (24–26 days), late-ripe (26–30 days), and very late-ripe (more than 30 days), which is consistent with the literature [27,28]. According to the previous investigations [27,28], there are four groups of ripeness in radishes: early-ripe (30–59 days), medium-early (60–69 days), medium-ripe (70–89 days), and late-ripe (90–120 days). In our study, precocious radish genotypes had a vegetative period of 30–41 days, medium-early 41–55 days, medium-ripe 55–69 days, and late-ripe 76–90 days. Most of the small radish and radish accessions belonged to the early-and mid-early ripening groups.

The beginning of root formation in small radish was observed on average at day 12–15 after shoot appearance. Radish root formation was observed on average at day 20–30. Based on the obtained data, it was revealed that the durations of vegetative phase for small radish and radish significantly differ. Small radish has a smaller range of variation. Statistically significant differences are found between European and Chinese forms. Among subsp. *sativus* accessions forms with a long root are more late-maturing than spherical or oval root forms. No significant differences were found between the types of Saxa, Scarlet globe, Pinkie, Pink-red with a White Tip, and French Breakfast. Accessions of these types may belong to different groups of ripeness.

Among the types of radishes, significant difference was found between convar. *hybernus* and convar. *sativus* with subsp. *sinensis* and subsp. *acanthiformis*, and among themselves. Accessions of Chinese radish var. *lobo*, var. *virens*, var. *syringeus*, var. *incarnatus*, and var. *rubidus* practically do not differ in the duration of the vegetative phase, except for accessions of the Ashkhabadskaya type. Accessions of this group have a later technical root ripening onset. The types of Japanese daikon radish do not have statistically significant differences in the vegetative phase duration among themselves.

In small radish, after the emergence of seedlings, leaves appeared on day 5–8, and in radish on day 8–10. By the beginning of marketable ripeness, small radish plants, depending on the accession, formed 4–6 leaves in subsp. *sativus* and 6–9 in subsp. *sinensis*. Radish plants, depending on the variety, formed 8–13 in subsp. *sativus*, 10–17 in subsp. *sinensis*, and 15–30 in subsp. *acanthiformis* leaves.

Accessions that form a large root crop with a relatively small photosynthetic apparatus are especially valuable.

Between the studied small radish accessions, the height of the rosette varied within 10.17–30.50 cm (CV = 19.7%), and the diameter of the rosette 11.58–21.72 cm (CV = 14.5%), while in radish these values were 14.28–57.75 cm and 21.00–57.30 cm (CV = 27.1% and 20.7%), respectively (Figure 2, Table 1).

Based on the data of the small radish rosette size, it was revealed that the small radish collection was statistically significantly divided into 11 groups according to the height of the rosette, and by the diameter of the rosette into six groups with a 2.0 cm interval (Figure 2). Accessions with medium-sized rosette (height and diameter 16.0–20.0 cm) were prevalent. The small size of the leaf rosette (up to 16 cm) was mainly characterized by accessions of the Saxa, Pinkie, and Scarlet globe types from Europe and North America, large rosettes formed accessions of the Chinese subspecies, as well as the Long Scarlet and White Icicle European small radish types.

For small radish, breeding of varieties with limited leaf growth and intensive development of root is relevant, so varieties with a small number of leaves and a compact rosette are of interest. As a result of the comparative study of small radish accessions of the European and Chinese subspecies, it was revealed that it is possible to select plants with a small rosette among the accessions of the European subspecies var. *rubescens* and var. *striatus*. As a result of our study, we found accessions of small radish with a stable manifestation of the desired traits of the leaf apparatus, which can be used as a starting material for breeding. Among the studied accessions, the following were particularly promising for breeding: accessions of the Saxa type from the Netherlands (‘KD’, k-2167; ‘Inca’, k-2202; ‘Revosa’, k-2228; ‘Bov’, k-2404; ‘Minitas’, k-2405; ‘Notar’, k-2408), Denmark (‘Saxa 455’, k-2154), Sweden (‘Saxa’, k-2299), Iceland (‘Saxa’, k-2343), and Tanzania (‘Cherry belle’, k-2133), type Scarlet globe from Canada (‘Early comet’, k-1936; ‘Cavalier bright scarlet’, k-1941), French Breakfast type from France (‘De Pontvil’, k-2197; ‘Flamboyant 5’, v.k.-3248) and Denmark (‘Safir’, k-2371), Pinkie type from the Netherlands (‘Pink beaty’, k-2242) and Russia (‘Perviy rozoviy’, v.k.-3188; ‘Pioner na griadke’, v.k.-3198), and Purple type from Russia (‘Viola’, v.k.-3161; ‘Amethyst’, v.k.-3189).

The radish collection was divided into nine groups by the height of the rosette, and by the diameter of the rosette into seven groups with a 5.0 cm interval (Figure 2). Most of the accessions of the radish collection were also characterized by the average size and shape of the leaves rosette. The shape differed from a spreading to semi-spreading form. (height 20.0–30.0 cm, diameter 25.0–35.0 cm). A large rosette of leaves was formed by accessions of convar. *hybernus* and several Japanese radish specimens from Japan: ‘Sakata Tenshiun’ (k-2112), ‘Natsu Sakari’ (k-2157), ‘Shinuchi Sobutori’ (k-2136), ‘Unzen shigatsu’ (k-2063). Accessions of Chinese and Japanese radishes with a small compact rosette of leaves included var. *lobo* from Vietnam (‘Bai cu’, k-2122) and South Korea (‘Euisungbanchung’, k-2153; ‘Iangsu’, k-2173), var. *incarnatus* ‘Rozoviy blesk Misato’ (v.k-3382, Russia), ‘Vnuchka’ (v.k-3383, Russia), and Japanese radishes from Japan (‘Horiyou’, k-2161; ‘Local’, k-2155; ‘Miyashige Oonaga’, k-2034; ‘Mijshige long pointed rooted’, k-2093; ‘All season cross’, k-2137; ‘Unsen-4-gatsu’, k-1946; ‘Akasuji’, k-2160) and Russia (‘Petersburg’, v.k-3337; ‘Sasha’, k-2142) were detected.

The length of the leaf blade of small radish accessions varied in the range of 6.33–15.20 cm (CV = 16.3%), width 4.33–9.90 cm (CV = 17.9%), and in radish these values were 9.30–28.00 cm and 6.75–16.46 cm (CV = 22.8% and 17.1%), respectively (Table 1). In general, the leaf size of the small radish and radish was characterized by an average degree of variability.

The small radish collection was divided into 10 groups with a 0.9 cm interval by the leaf length, and into seven groups with a 0.8 cm interval by the leaf width. Most of the small radish accessions were characterized by the following leaf sizes: length 9.8–12.5 cm, width 5.9–8.3 cm. The radish collection was divided into eight groups with a 2.3 cm interval by the leaf length, and into 11 groups with a 0.8 cm interval according to the leaf width. Large leaves with a broad leaf blade were formed mainly by plants of var. *rubidus* with a rounded root and by some Japanese radishes of the Minovase type. The same morphological type is characteristic for the small radish accessions of subsp. *sinensis* and Long Scarlet types.

In the studied set of small radish and radish, there was a variety in shape (entire, sinuate, lyrate), color (dark green, green, yellow-green), and pubescence of leaves. Most of the specimens had lyrate leaves with varying degrees of pubescence, from weak to strong. Local small radish accessions from Burundi (k-2424, type White Icicle) and subsp. *sinensis* accessions with red (k-1176, China; k-1667, Russia; k-2187, Azerbaijan; k-1946, Tajikistan; k-1233, China) and white (k-1666, Russia; k-1921, China; k-1923, China), color of the root had an entire, unpeeled leaf plate, i.e., high-quality “salad-type” leaves.

The length of the root in the studied accessions of small radish varied in the range of 2.34–13.83 cm (CV = 46.7%), the diameter 1.41–5.36 cm (CV = 19.8%), and the root index (length/diameter) 0.81–9.82 (CV = 71.6%). The small radish accessions were divided into 11 groups with 1.1 cm interval. The length of the root, and root diameter were divided into 11 groups with 0.4 cm interval and by the root index into nine groups with a 0.6 interval. The majority of the studied accessions was characterized by the root length from 2.34 to 4.40 cm and a diameter from 2.12 to 3.21 cm. These accessions had a rounded or oval shape of the root (root index 0.81–1.41). Varieties with a round and oval root crop with a diameter of at least 2.5 cm are mainly used for breeding.

Among the radish accessions, a wide variety was also noted in root length (5.30–37.64 cm, CV = 48.6%), root diameter (2.70–8.95 cm, CV = 21.9%), and shape (0.80–10.24, CV = 65.2%). The collection was divided according to the length of the root into seven groups with a step of 4.6 cm, by diameter-into eight groups with a 0.8 cm interval and by the root index into eight groups with a 1.2 cm interval. Most of the studied accessions were characterized by the length of the root crop from 9.92 to 19.16 cm and the diameter from 5.04 to 6.61 cm, with an oval or cylindrical shape of the root crop.

The following forms of roots in the studied set of *R. sativus* varieties were observed: flat or transverse elliptic (shape index less than 1.0), round (1.0–1.20), elliptic (1.21–2.0), cylindric and elongated-cylindric (2.01–4.0), conical and fusiform (4.01–6.0).

The studied accessions of *R. sativus* had different color variants of the bark and root pulp. The small radish had white, pink, pink-red, red, crimson, purple, yellow bark color, as well as two-color samples (with a white root tip). The radish had a greater variety in the color of the bark, namely white, white with a green color in the upper part of the root, black, purple, green, green with a white tip, pink, pink-red, and red. The root pulp of the small radish and radish accessions was mostly white, but there were radish accessions with light green (var. *virens*), red (var. *incarnates*) and purple (var. *syringeus*) pulp color.

The variability of plant weight in small radish and radish (17.69–114.50 g and 113.40–1370.50 g) and the root weight (10.22–75.20 g and 69.00–1057.19 g) was high (CV over 30%). According to the root weight in the small radish, nine groups were statistically significantly different with a 5.0 g interval (Figure 3). A small root mass (up to 15.0 g) in small radish was associated to types with a round and elliptic root and in some accessions with a cylindrical root. The majority of the accessions had a root weight of 15.0–25.0 g. This group included accessions belonging to var. *rubescens*, var. *striatus*, var. *violaceus*, and var. *chloris* with a rounded and elliptic shape of the root crop, as well as with an elliptic and elongated-cylindrical shape (var. *striatus* and var. *radicula*). The weight of 25.0–30.0 g was found in accessions of the types White Icicle, Long Scarlet and several accessions of the type Saxa with a root diameter of 3.0–3.2 cm, as well as accessions of subsp. *sinensis* with a long root. A weight greater than 30 g was typical for accessions of subsp. *sinensis* with a long and round root, and for several accessions of subsp. *sativus*: type Pink-red with a White Tip (‘Rund halbrot halbweiss’, k-2375, Denmark), type Yellow (‘Zolotce’, v.k. -3342, Russia), type Pinkie (‘XXL’, v.k.-3229, Russia), and Hailstone (‘White radish’, v.k.-3446, Russia).

The radish collection was divided into four groups with a 150 g interval by the mass of the root. The smallest root crop masses (up to 200 g) was mainly found in accessions of convar. *sativus*. The main part of the accessions of the collection had a weight of 200–350 g. The weight of more than 350 g was found in accessions of convar. *hybernus* and Minovase type Japan radishes (Japanese accessions ‘Natsu Salari’, k-2157; ‘F1 Minowase Summer gross No. 2’, k-2228; accessions from China (v.k-3313, v.k-3314, v.k-3315) and Miyashige (‘Kioba Kairio Kameida’, k-2113, Japan; ‘Haruyosi 260’, k-2127, Japan).

Accessions with a high mass of the root crop can be used as a starting material in breeding for high productivity.

An important feature that determines the crop value is the market quality of the root. The main reasons for decrease in the market quality of root crops are non-simultaneous root formation, early bolting, cracking, formation of unaesthetic roots and voids inside them. In our study, the market quality varied from 10% to 95%. The smallest number of marketable roots in small radish was formed by Long Scarlet type accessions, because of non-simultaneous root formation and early bolting of plants. The French Breakfast type accessions showed rapid flabbiness of the pulp after they reached technical ripeness, which significantly reduced their market quality. Round accessions of var. *rubescens* had a large range of market quality, from 50% to 95%. Accessions of subsp. *sinensis* had a dense juicy pulp, retained their marketability for a long time, and had no tendency for early bolting.

Accessions of radish with low market quality (less than 50%) were represented by local accessions of Chinese radish from China, South Korea, and Kyrgyzstan and several accessions of daikon (convar. *acanthiformis*). High market quality (more than 80%) was noted in convar. *hybernus* accessions and some Japanese daikon of the Minovase, Miyashige, Ninengo, Shiroagiri types. Among the other accessions, there was a significant variability in this trait, which is determined by the characteristics of the genotype. It is noted that the early bolting of radish has a slight effect on the quality of roots. Specifically, it continues to grow and develop simultaneous with peduncle formation and without deterioration in the quality of the pulp.

Thus, among the quantitative characteristics of small radish and radish, high variability is noted for the characteristics of the roots (length, root index, and weight). At the same time, radish has a higher variation for all the studied traits, which indicates a greater genetic diversity. In general, the differences between the types of small radish regarding the main morphological characteristics of the leaf and root crop and productivity are clearly manifested at the level of subspecies: subsp. *sativus* and subsp. *sinensis*, as well as between var. *rubescens*, var. *striatus*, var. *violaceus,* and var. *radicula*. The Long Scarlet type, which belongs to var. *rubescens*, is significantly different in all the studied traits from other types of this variety. In radish accessions, there were no significant differences between the types based on the morphological characteristics of the rosette and leaf and productivity, while significant differences between the types were found based on the characteristics of the root crop.

### 2.2. Variability of Biochemical Characteristics

The value of small radish and radish for the human consumption is determined by the content of the main nutrients, as well as by presence and amount of biologically active substances. Species and varietal characteristics, weather conditions, place of cultivation, and the phenological phase affect the concentration of chemical components [28]. An important indicator that characterizes the nutritional value of vegetables is the content of dry matter, which determines the yield of biomass.

In our study, the overall range of dry matter content variability in root crops was quite high. The dry matter content in the root crops ranged from 3.72% to 8.56% in the small radish (CV = 21.4%), and from 5.28% to 13.64% in the radish (CV = 24.1%) (Figure 4, Table 2). At the same time, the accessions of the small radish subsp. *sinensis* accumulated more dry matter than the accessions of subsp. *sativus*, and in the radish, on the contrary, the accessions of subsp. *sativus* were characterized by high dry matter content.

According to the dry matter content in the roots of small radish, six groups were identified with a 1.0% interval. Most of the studied accessions of small radish had a dry matter content in the range of 4.0–7.0%. A small percentage of dry matter (up to 4.0%) was typical for precocious accessions of the Saxa type, and the highest amounts were found in (more than 7%) accessions of subsp. *sinensis*, French Breakfast type from Sweden (‘Delikat’, k-2176,) and var. *radicula* from Russia (‘Aphrodite’, v.k-3212) and Burundi (‘Local’, k-2424).

According to the dry matter content, the radish collection was reliably divided into six groups with a 1.5% interval. The majority of radish accessions accumulated dry matter in the range of 6.5–9.5%. A small percentage of dry matter (up to 6.5%) was observed in local accessions of var. *lobo* from Kyrgyzstan and Egypt, as well as in accessions of Chinese radish from Chile and Mongolia and two accessions of Japanese radish of the Kameido type (‘Eifuku 2’, k-2134) and Minovase (‘Natsu Salari’, k-2157). All convar. *hybernus* accessions were characterized by a high dry matter content (more than 9.5%), and several accessions of Chinese radish (‘Rubinoviy surprise’, v.k-3139, ‘Saharnaya roza’, v. k-3284, ‘Red meat’, v.k-3381) were also distinguishable in this trait.

Accessions of small radish and radish with a high dry matter content can be used as sources for breeding for high dry matter content.

About 80% of the dry matter consists of carbohydrates such as sugars, starch, fiber, pectin, and other substances. The predominant part of the dry matter of small radish and radish root crops is represented by mono-and disaccharides, and therefore this indicator is of great importance for the comparative evaluation of accessions. Depending on the variety, the sugar content in the dry matter can reach 25–55% [27,29].

In our studies, the sugar content in the dry matter was 35–77% in small radish, and 13–47% in radish. The high variability of this biochemical trait is associated with both genetic characteristics and climatic conditions of cultivation. The total sugar content averaged 2.14% (0.16–5.45%) in small radish and 2.34% (1.03–4.31%) in radish, of which the monosaccharide content was 1.9% and 2.0%, respectively.

In small radish, seven groups with a 0.76% interval have been identified according to the sugar content. Low sugar content (less than 1.0%) was observed in most early ripening small radish types Saxa, Pink-red with a White Tip and French Breakfast. A high sugar content (more than 4.0%) was noted in the following accessions: var. *striatus*—‘French Breakfast’ (k-1939, Pakistan); var. *radicula* ‘Untitled’ (k-2379, Lebanon), ‘Syla’ (k-2347, Denmark), ‘Local’ (k-2424, Burundi); var. *rubescens* ‘Saratovsky’ (k-2210, Russia), and subsp. *Sinensis* ‘Darozi surkh local’ (k-1946, Tajikistan), ‘Local’ (k-2260, Russia), ‘Local’ (k-1923, China), and ‘Virovsky beliy’ (k-1666, Russia). The rest of the studied accessions accumulated sugar in the range of 1.00–3.18%

According to the sugar content in the roots, the radish collection was divided into five groups with a step of 0.7%. Most of the accessions were characterized by a content in the range of 1.0–3.0%, more than 3.0% accumulated in accessions of var. *lobo* (‘Local’, k-1978, Kyrgyzstan; ‘Chinese white winter’, k-2101, Chile; ‘Bai cu’, k-2122, Vietnam; ‘Euisungbanchun’, k-2153, South Korea), var. *virens* (‘Zelenaya boginya’, v.k.-3140, Russia), var. *incarnatus* (‘Rubinoviy surpriz’, v.k.-3139, Russia; ‘Saharnaya roza’, vr.k-3284, Russia; ‘Red meat’, v.k.-3381, Russia) and var. *syringeus* (‘Da 8’, v.k.-2899, Russia), Japanese radishes (‘F1 Minowase Summer gross No. 2’, k-2228, Japan; v.k -3314, China, v.k -3324, China), and convar. *hybernus* (‘Autumn Gurnay’, k-1614, Germany).

Ascorbic acid (vitamin C) is one of the important biologically active substances in *R. sativus* crops. In the roots of small radish and radish, it is present in its free form, bioavailable to humans. The average content of ascorbic acid in the roots of small radish is 30.47 mg/100 g (19.04–57.72 mg/100 g), and in radish 42.02 mg/100 g (11.80–73.32 mg/100 g) (Figure 5, Table 2).

The amplitude of ascorbic acid variability in small radish roots is average (CV = 20.6%), the studied accessions were divided into eight groups with a 5.0 mg/100 g interval. Smallest contents (up to 25.0 mg/100 g) were observed in medium-early accessions of var. *striatus*, var. *rubescens* and var. *violaceus*, as well as in single accessions of var. *radicula* and var. *chloris*. More than 35.0 mg/100 g of ascorbic acid were accumulated by accessions of the Saxa type from the Netherlands ‘Revosa’ (k-2228) and ‘Neoro’ (k-2166), from Bulgaria ‘Lubimi’ (k-2219), Moldova ‘Giocel’ (k-2325), Iceland ‘Saxa’ (k-2343), type French Breakfast (‘Flamboyant 5’, v.k-3248, France), and Hailstone (‘Aphrodite’, v. k-3212, Russia).

The variability of ascorbic acid content in radish root crops was significantly higher (CV = 37.2%) than in small radish. The collection was divided into six groups in increments of 12.0 mg/100 g. Low ascorbic acid content (less than 23.0 mg/100 g) was observed in some accessions of convar. *hybernus*, var. *lobo*, and Japanese radish (‘Kono Hayabutori’, k-2110, Japan). The majority of the accessions were characterized by a content in the range of 35.0–59.0 mg/100 g. Accessions of Japanese radish from Japan (‘Unzen shigatsu’, k-2063, Japan; ‘Mijshige long pointed rooted’, k-2093, Japan), Chinese radish var. *virens* (‘Wei-xiang’, k-1865, China), var. *lobo* (‘Local’, k-1978, Kyrgyzstan), and var. *rubidus* (‘Nejnaya’, k-1983, Russia) accumulated 62.0 mg/100 g of ascorbic acid. The highest contents were observed in accessions of Chinese radish var. *rubidus* ‘No. 129’ (k-2100, Mongolia) and ‘Red ballof changchou’ (k-1906, China) with 67.7 and 73.3 mg/100 g, respectively.

We have identified significant differences in the content of dry matter and sugars between the subspecies of radish and small radish, and differences in the content of dry matter and ascorbic acid between two radish crops.

### 2.3. Principal Component Analysis (PCA)

In breeding, the correlating inheritance of leaf, rosette, and root traits, as well as the content of the main biochemical components, are of interest.

Based on the evaluation results, 128 accessions of *R. sativus* were grouped by geographical origin. The countries of origin of the accessions were divided into groups according to the geographical proximity. Accessions from Russia, China, Japan, and South Korea were separated into separate groups.

As a result of the analysis of 12 phenotypical and biochemical traits according to the method of principal component analysis (PCA), it was found that their variability is determined by four factors. Together, they determine 91.51% of the total variance by. At the same time, the first component that characterizes the parameters of the rosette and the leaf (the height and diameter of the rosette, the length and width of the leaf), the diameter of the root, and the weight of the plant and the root determines 61.32% (Table 3).

The second component determines the length and index of the root. The third determines the content of ascorbic acid, while the fourth determines the content of dry matter and the duration of the vegetative phase.

Thus, the analysis demonstrates the relationship between the characteristics of the leaf and the rosette with the signs of productivity. The length of the root and index of the root is not related to the diameter. The dry matter content is associated with the duration of the vegetative phase, which implies that, with an increase in the maturation period, more dry matter accumulates in root crops.

Since the first two components characterize the largest part of the variability of features, we considered the location of accessions in their space (Figure 6).

As a result, the accessions were divided into three groups (Figure 6). The first group includes 72 small radish accessions and 39 radish accessions, which are characterized by a short duration of the vegetative phase, medium rosette size, a root shape from rounded to cylindrical, a small root weight, high ascorbic acid content, and low dry matter content. These are mainly accessions of small radishes from Europe, Russia, USA, and Canada, Chinese radishes of the Chinto Lobe type from South Korea, Western Asia, and Chile, while accessions type Hun-dong-lon of various origins and daikon from Japan are also included in this group.

The second group includes two small radish accessions and 16 radish accessions. The accessions are characterized by late maturity, large size of the leaves and rosette, a significant root weight and a high content of dry matter and ascorbic acid located in the roots. This group is mainly represented by accessions of radishes of the European subspecies, such as Round Black Spanish, Winter white round, and radishes of the Chinese subspecies, such as Xing-li-mei with red flesh.

The accessions of the third group were characterized by a large variability in the combination of the studied features and formed a rather large and sparse cloud in the factor space. This group includes 33 accessions of small radish of various types (mainly Icicle and Purple types) from Africa, Northern and Western Europe, the USA, Chile, and Russia. The group also included 56 accessions of radish of European, Chinese, and Japanese subspecies of various types.

Separately from the groups, there are accessions of small radish and radish with extreme values of individual characteristics that exceed the specified limits of these characteristics for the groups to which they belong.

## 3. Discussion

Genetic diversity is the basis of biological diversity which determines the ability of a species to adapt to the external environment and evolve [30]. A lower genetic diversity of a species means less genetic variation and adaptability, which threatens its long-term survival [31,32]. The role of world genetic resources for economic use has been repeatedly emphasized by many researchers [33,34,35,36]. When creating new cultivars, the leading role belongs to the initial material for hybridization. Plant genetic resources have material and intellectual value, ensuring food security and economic development [37,38]. The collection, study, and preservation of genetic resources is the primary task of genebanks [39].

One of the strategies to study and preserve genetic resources is to create trait (phenotypic) and core collections [40,41,42]. Trait collections are formed using phenotypically different forms. They are created on the basis of species and varietal collections stored in genebanks or research institutes. The basis of such collections can be a set of cultivars with clearly defined traits of interest to the researcher. Trait collections contain accessions that contrast in quantitative and qualitative characteristics, i.e., with a well-expressed alternative value of the traits [40,43,44]. A core collection is a subset of a large germplasm collection that contains accessions chosen to represent the genetic variability of the germplasm collection. The purpose of the core collection is to improve the management and use of a germplasm collection. Core collections are usually assembled by grouping accessions and selecting from within these groups. Core collections are of strategic importance as they allow the use of a small part of a germplasm collection that is representative of the total collection [45,46].

Genetic variability is of paramount importance in selecting the best genotypes for making rapid improvement in yield and desirable characters as well as for selection of the most promising parents for further breeding program. Study of genetic variability reveals variation in different quantitative and quality traits. Moreover, genetic variability represents an essential and basic requirement for crop improvement as it provides wider scope for selection when initiating an effective and successful breeding program [47].

During the long history of radish cultivation, humans have produced a great diversity of *R. sativus* morphological types. Only a few studies have been conducted to estimate the phenotypic diversity of radish varieties [34,47,48,49,50]. None of the currently available studies has focused on a wider set of European and Chinese radish and small radish varieties and thus may not provide a sufficient understanding of the genetic diversity of *R. sativus*. In addition, selection indices for the production breeding of these crops have not yet been improved and the available information is meager and inadequate.

This study confirmed that the small radish and radish differ greatly in morphological, biochemical and economically valuable characteristics of the plant at the level of subspecies, groups of varieties, varieties and types. Accessions within the same type are often difficult to distinguish by external features. In our work, the previously described limits of variability of important morphological, biochemical, and economically valuable traits have been expanded, which is of particular value when choosing a source material for breeding these crops.

In recent years, *R. sativus* crops have been actively studied using molecular markers. Many studies have noted the effectiveness and prospects of using molecular markers AFLP, RAPD, and ISSR, as well as some biochemical markers for assessing the genetic variability of *R. sativus* [51,52,53,54,55,56] and creating core collections [57]. Our present study and previous work [58,59] indicate the need to use morphological and biochemical markers to study intraspecific variability. A consequence comprehensive study of the genetic diversity of *R. sativus* will make it possible to more clearly determine the course of the morphogenetic process and phylogenetic relationships within the species.

The Department of Vegetable crops and Cucurbits of the VIR is working on the creation of trait and core collections based on the extensive collection of *R. sativus* at the Institute [60,61]. The collection includes accessions that differ in the length of the duration of vegetative phase, in morphological and phenotypical characteristics, in biochemical composition, and in resistance to abiotic environmental factors [62,63,64]. When forming a trait collection, the genetic origin of the accessions, as well as botanical, agricultural, and biological characteristics are considered. Every year, the world collection is screened to identify both new phenotypes for individual traits and combinations of already known traits, which allows us to expand the understanding of the genetic diversity of the species. Further, our trait collections can serve as a basis for the formation of the core collection of *R. sativus*. Currently, we have described most of the collection according to the minimum descriptor, including 30% of the collection studied for 30 phenotypic characters in various growing conditions [63]. We study the detailed chemical composition of leaves and roots using metabolomic profiling. We also analyzed part of the accessions with a set of molecular markers [65]. These accessions are included in the core collection.

Our investigation concerning the degree of variation regarding phenotypical indicators allowed us to identify adaptive stable and highly variable traits and properties of *R. sativus* accessions. This is necessary for the creation and use of trait (phenotypic) collections, which is the most effective way one of the possible approaches to structure collections of plant genetic resources of any crop.

## 4. Materials and Methods

### 4.1. Plant Material

The research work was carried out in the Vavilov Federal State Scientific Research Institute of the Russian Academy of Sciences (VIR) at the Pushkin and Pavlovsk Laboratories of VIR (Pushkin, Leningrad Region) in 2016–2020.

The research material was a set of accessions of small radish and radish from the VIR collection of various ecological and geographical origin, fully covering the diversity of the species. Accessions of different ecological and geographical origin, almost completely represent the diversity of the species. The studied small radish sub-collection was represented by 149 accessions from 37 countries, belonging to 13 types of 7 varieties of European and Chinese subspecies. The radish sub-collection was represented by 129 accessions from 21 countries, belonging to 18 types of 11 varieties of European, Chinese and Japanese subspecies. This set represents the global diversity of cultivated forms of small radish and radish, and also includes local varieties from the centers of origin of these crops.

### 4.2. Field Evaluation and Agronomic Practices

Field studies were conducted in accordance with the “VIR Guidelines for the study and maintenance of the world collection of root crops” (1989) [66], “Guidelines for the testing of vegetable and Melon crops” (2018) [67].

Small radish: Sowing in the field was carried out in mid-May, the area of the plot was 1 m^2^. The planting scheme is 10 × 5 cm, in two repetitions. The light period in May is 16.0–18.2 hours, and in June 18.3–18.9 hours, provoking an early bolting. The average temperature in May during the day is 12.4–17.5 °C and at night 5.2–8.7 °C, while in June during the day it is 15.6–19.2 °C and at night 10.3–13.5 °C. Humidity of the air was 65–70%, and soil humidity 5–80%.

Radish: Sowing in the field was carried out in mid-July. The area of the plots is 1.4 m^2^, the planting scheme is 70 × 15 cm, in two repetitions. The length of the day from July to September decreased from 18.7 to 12.3 hours. The average temperature in July in the afternoon is 21.9–23.4 °C and at night−15.1–16.4 °C, in August in the afternoon 18.6–21.2 °C and at night 14.3–15.5 °C, in September in the afternoon 17.6–19.2 °C and at night 12.6–14.5 °C. Air humidity is 70–75% and soil humidity 80–85%.

During the growing season, phenological observations, biometric measurements, and morphological description of plants were carried out. The following parameters were taken into account: the shape of the rosette, the height and diameter of the rosette, the number of leaves, the color of the leaf blade, the shape of the leaf blade, the length and width of the leaf blade, the length and diameter of the root, the shape of the root, the color of the root skin and inner part, the weight of the root, the ratio of the mass of the root to the mass of the plant, yield, and marketability.

### 4.3. Biochemical Analysis

The biochemical analysis was carried out in the Department of Biochemistry and Molecular Biology of the VIR in the phase of technical ripeness of root.

The analysis and processing of the material was carried out according to the method of the Department of Biochemistry of the VIR [68]: the dry matter mass content was determined by weighing before and after drying the average accession in a FED 400 Binder thermostat (Germany) at 105 °C; sugars by the Bertrand method [69]; ascorbic acid by direct extraction from plants with 1% hydrochloric acid, followed by titration with 2,6-dichlorindofinol (Tilman’s reagent) [70]. All data are provided in terms of raw material.

### 4.4. Statistical Analysis

Data analysis was performed using the software STATISTICA v. 12. 0 (StatSoft Inc., Tulsa, OK, USA). Descriptive statistics (mean, standard deviation, range of variability, coefficient of variation) were calculated for all parameters. Data testing for the normality of the distribution was performed using the Shapiro–Wilk test and the quantile-quantile graph (QQ Plot). The average values of data with a normal distribution were compared using one-factor analysis of variance (ANOVA), while data with a distribution other than normal were compared using the Kruskal–Wallis test. The Tukey HSD (honestly significant difference) test was used to identify the differences. The variability of the structure of feature relationships was evaluated using principal component analysis (PCA). Factor loads were expressed in correlation coefficients with the factor. In addition, the eigenvalues for each factor, the share of factors in the total variance, and the cumulative share of extracted factors were calculated. The selection of the number of optimal factors was carried out using the Scree test [71].

## 5. Conclusions

As a result of the evaluation of *R. sativus* accessions according to phenological, morphological and biochemical analyses, a wide variation of these characteristics was revealed, which is due to the large genetic diversity of small radish and radish of different ecological and geographical origin. 

The variability of the duration of vegetative phase, the main morphological features of the leaf rosette and root, the traits of productivity, and the components of the chemical composition of the root in a representative set of accessions of the small radish and radish VIR collection were revealed. Statistically significant differences were found between the European, Chinese, and Japanese forms of small radish and radish. It was found that the greatest amplitude of variation in small radish referred to the morphological characteristics, such as length and mass of the root. In radish, an average variability was revealed by the diameter of the rosette and the width of the leaf. Other characteristics of the leaf apparatus and the root have a significant degree of variability. Significant differences in the content of dry matter and sugars between the subspecies of small radish and radish, and differences in the content of dry matter and ascorbic acid between different groups of the two crops were revealed. The values of phenotypical and biochemical traits in the descriptor for small radish and radish were revised and refined. The sources of valuable breeding traits for earliness, compact rosette, high productivity, as well as high content of dry matter and ascorbic acid were identified in the VIR radish collection.

## Figures and Tables

**Figure 1 plants-10-01799-f001:**
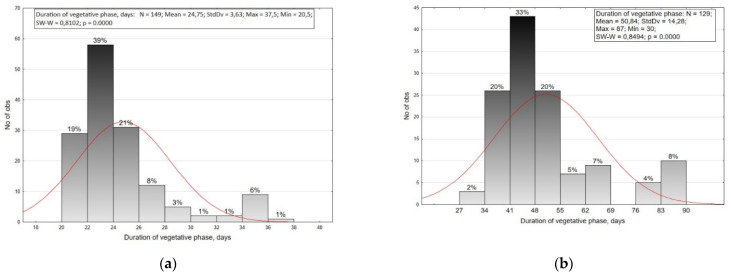
Distribution of small radish (**a**) and radish (**b**) accessions by the duration of vegetative phase.

**Figure 2 plants-10-01799-f002:**
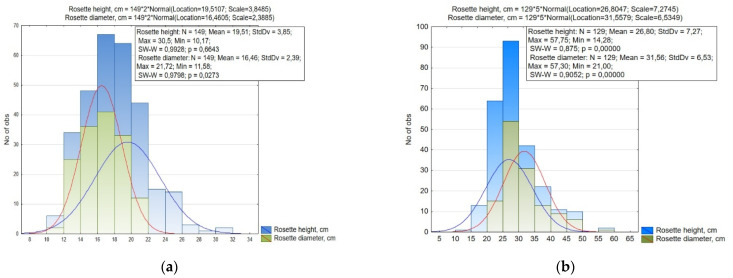
Histogram of the distribution of (**a**) small radish and (**b**) radish accessions by rosette size (height and diameter).

**Figure 3 plants-10-01799-f003:**
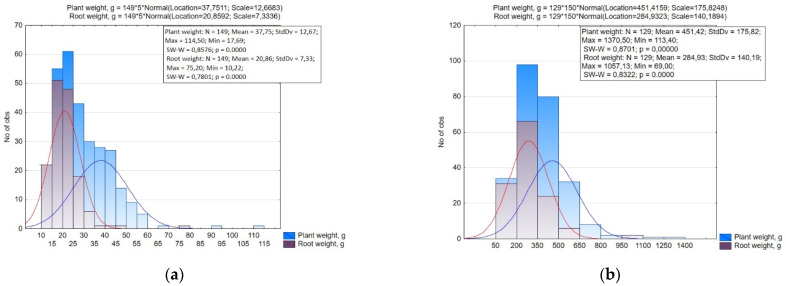
Histogram of the distribution of accessions of small radish (**a**) and radish (**b**) by the weight of the entire plant and root.

**Figure 4 plants-10-01799-f004:**
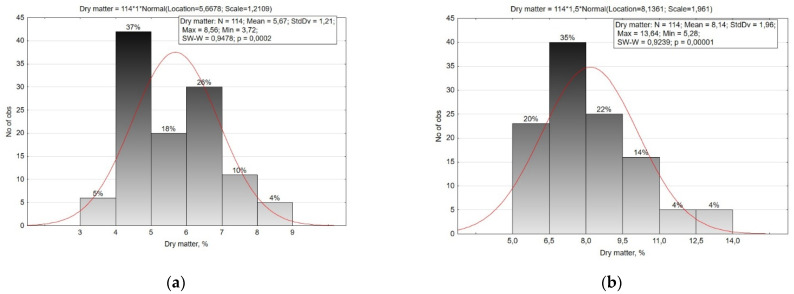
Histogram of the distribution of small radish (**a**) and radish (**b**) accessions by the content of dry matter in the roots.

**Figure 5 plants-10-01799-f005:**
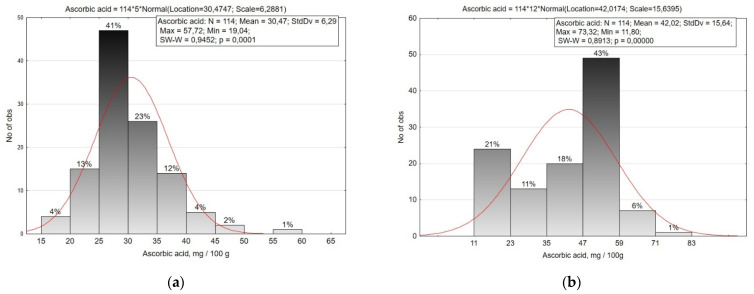
Histogram of the distribution of small radish (**a**) and radish (**b**) accessions by the content of ascorbic acid in the roots.

**Figure 6 plants-10-01799-f006:**
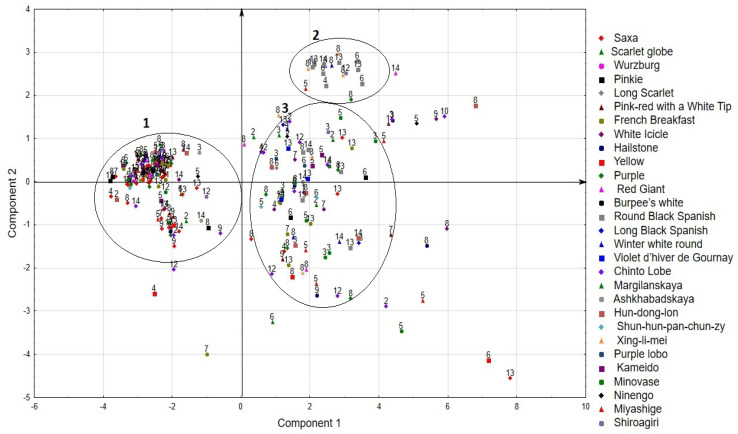
Distribution of accessions in the space of the first two components. The color indicates the types of accessions. The numbers of the group according to the origin of the accessions are: 1—Africa, 2—Central Asia, 3—China, 4—East Europe, 5—Japan, 6—North America, 7—North Europe, 8—Russia, 9—South America, 10—South Asia, 11—South Europe, 12—South Korea, 13—West Europe, 14—Western Asia.

**Table 1 plants-10-01799-t001:** Characteristics of small radish and radish variety types by morphological characteristics and productivity, average for 2016–2020.

Type (Number of Accessions)	Duration of Vegetative Phase. Days	Rosette Height. cm	Rosette Diameter. cm	Leaf Length. cm	Leaf Width. cm	Root Length. cm	Root Diameter. cm	Root Index (Length/Diameter)	Plant Weight. g	Root Weight. g
*R. sativus* subsp. *sativus* convar. *radicula* (Pers.) Sazon.										
var. *rubescens* Sinsk.										
Saxa (44)	22.54 ± 1.30 ^agj^	18.82 ± 3.34 ^bdefj^	15.06 ± 1.52 ^bdf^	10.88 ± 1.65 ^acdegj^	6.77 ± 1.09 ^aef^	3.32 ± 0.40 ^a^	3.01 ± 0.30 ^a^	1.11 ± 0.14 ^a^	35.48 ± 8.79 ^abd^	19.69 ± 4.40 ^abcd^
	20.50–25.80	11.10–26.30	12.80–18.70	7.50–15.00	4.60–9.30	2.30–4.30	2.20–3.50	0.90–1.50	17.70–50.10	10.20–28.10
	5.80%	17.80%	10.10%	15.20%	16.10%	12.00%	10.10%	12.60%	24.80%	22.40%
Scarlet globe (5)	22.90 ± 0.74 ^abegj^	17.50 ± 6.09 ^bdefj^	14.24 ± 1.98 ^bdf^	9.72 ± 2.87 ^bdef^	6.00 ± 1.51 ^bdf^	3.58 ± 0.36 ^a^	2.86 ± 0.40 ^a^	1.28 ± 0.26 ^a^	29.94 ± 8.95 ^ab^	15.82 ± 2.94 ^ad^
	21.80–23.50	11.30–25.60	11.90–16.40	6.90–13.80	4.40–7.70	3.20–3.90	2.30–3.20	1.00–1.70	19.20–43.00	12.40–20.20
	3.20%	34.80%	13.90%	29.50%	25.10%	10.00%	13.90%	20.20%	29.90%	18.60%
Würzburg (4)	25.48 ± 0.45 ^bdef^	16.89 ± 2.36 ^bdefj^	14.82 ± 1.28 ^bdf^	10.77 ± 1.17 ^acdegj^	7.35 ± 0.92 ^aeg^	3.53 ± 0.57 ^a^	3.07 ± 0.40 ^a^	1.16 ± 0.14 ^a^	36.33 ± 4.68 ^abcd^	19.93 ± 3.43 ^abcd^
	24.83–25.83	14.32–19.90	13.47–16.28	9.47–11.94	6.14–8.36	2.72–4.06	2.76–3.65	0.99–1.33	31.43–41.55	16.23–23.22
	1.80%	14.0%	8.70%	10.90%	12.60%	16.20%	13.20%	12.20%	12.90%	17.20%
Pinkie (7)	24.36 ± 1.14 ^ab^	15.60 ± 2.44 ^b^	14.33 ± 1.73 ^b^	8.79 ± 1.20 ^b^	5.21 ± 0.63 ^b^	3.31 ± 0.35 ^a^	3.00 ± 0.39 ^a^	1.11 ± 0.13 ^a^	32.43 ± 14.27 ^ab^	20.75 ± 9.69 ^ac^
	23.00–26.50	12.98–18.80	12.18–17.00	7.40–10.21	4.40–6.21	2.92–3.81	2.50–3.71	1.01–1.40	21.00–58.88	13.80–40.79
	4.70%	15.70%	12.10%	13.60%	12.00%	10.60%	13.10%	12.00%	44.00%	46.70%
Long Scarlet (6)	27.83 ± 1.46 ^ch^	27.01 ± 3.44 ^h^	20.84 ± 1.30 ^h^	13.61 ± 1.56 ^h^	8.95 ± 0.86 ^h^	9.67 ± 2.13 ^d^	2.10 ± 0.44 ^b^	4.99 ± 2.41 ^d^	53.31 ± 9.54 ^e^	26.64 ± 6.34 ^ce^
	25.50–29.80	22.05–30.50	18.23–21.70	11.50–15.10	7.48–9.90	7.90–13.83	1.41–2.80	3.30–9.82	42.00–69.20	20.40–37.60
	5.30%	12.70%	6.30%	11.50%	9.70%	22.10%	21.10%	48.30%	17.90%	6.30%
var. *striatus* Sinsk.										
Pink-red with a White Tip (18)	23.38 ± 1.13 ^a^	19.94 ± 4.06 ^a^	16.91 ± 2.22 ^a^	10.59 ± 1.73 ^a^	6.83 ± 1.23 ^a^	3.24 ± 0.60 ^a^	3.00 ± 0.48 ^a^	1.14 ± 0.42 ^a^	34.54 ± 9.64 ^a^	19.28 ± 5.36 ^ac^
	21.30–25.00	10.17–27.30	11.58–19.90	6.33–12.90	4.33–8.90	2.50–4.96	1.93–3.90	0.81–2.57	20.10–52.50	10.70–32.00
	4.80%	20.40%	13.10%	16.30%	17.90%	18.50%	16.20%	36.50%	27.90%	27.80%
French Breakfast (17)	22.28 ± 1.04 ^g^	21.76 ± 3.04 ^acg^	17.52 ± 1.54 ^aeg^	11.23 ± 1.31 ^aceg^	7.42 ± 0.89 ^aceg^	5.61 ± 0.90 ^c^	2.20 ± 0.23 ^b^	2.59 ± 0.53 ^c^	33.54 ± 7.50 ^abd^	17.86 ± 3.34 ^acd^
	20.50–24.08	12.83–25.53	13.33–19.56	7.33–13.06	4.83–8.88	4.25–7.92	1.87–2.69	1.66–3.84	19.99–43.94	11.09–25.03
	4.70%	14.00%	8.80%	11.60%	12.00%	16.10%	10.70%	20.40%	22.30%	18.70%
var. *radicula*										
White Icicle (12)	26.83 ± 2.06 ^c^	20.24 ± 3.15 ^ac^	18.34 ± 2.38 ^c^	10.83 ± 1.33 ^ac^	7.38 ± 1.24 ^ac^	7.71 ± 1.69 ^b^	2.09 ± 0.19 ^b^	3.78 ± 1.10 ^b^	40.07 ± 8.52 ^ac^	21.36 ± 3.64 ^abc^
	22.50–30.50	14.17–25.80	13.28–21.30	8.44–13.20	5.10–9.10	5.30–11.56	1.80–2.40	2.20–6.19	25.50–52.50	14.50–26.96
	7.70%	15.60%	13.00%	12.30%	16.80%	22.00%	9.20%	29.20%	21.30%	17.00%
Hailstone (6)	24.22 ± 1.00 ^abde^	17.34 ± 2.29 ^bde^	17.52 ± 2.48 ^acde^	10.92 ± 1.10 ^abcde^	6.84 ± 0.82 ^ace^	3.69 ± 1.09 ^a^	3.31 ± 1.08 ^a^	1.13 ± 0.09 ^a^	47.16 ± 23.69 ^acd^	27.28 ± 14.14 ^abc^
	23.00–26.00	14.90–21.10	14.10–20.20	9.90–13.00	6.00–7.83	2.75–5.75	2.30–5.36	1.00–1.26	26.33–114.50	11.83–75.20
	4.10%	13.20%	14.20%	10.10%	12.10%	29.40%	32.80%	8.20%	41.50%	38.50%
var. *chloris* Alef.										
Yellow (5)	25.86 ± 0.91 ^bcdef^	16.81 ± 1.06 ^bdef^	14.72 ± 1.23 ^bdf^	9.24 ± 1.00 ^bdf^	5.96 ± 0.43 ^bdef^	3.46 ± 0.45 ^a^	3.08 ± 0.29 ^a^	1.13 ± 0.07 ^a^	34.80 ± 7.70 ^abcd^	21.74 ± 7.20 ^abc^
	24.50–26.80	15.30–18.10	12.83–15.80	8.00–10.71	5.48–6.50	3.20–4.25	2.87–3.54	1.03–1.20	26.90–45.74	14.00–33.38
	3.50%	6.30%	8.40%	10.80%	7.30%	12.90%	9.30%	6.20%	22.10%	33.10%
var. *violaceus*										
Purple (12)	24.79 ± 0.84 ^bd^	17.88 ± 2.68 ^bd^	15.89 ± 1.58 ^bd^	10.13 ± 0.70 ^abcd^	5.98 ± 0.79 ^bd^	3.31 ± 0.35 ^a^	2.93 ± 0.27 ^a^	1.13 ± 0.11 ^a^	35.63 ± 6.40 ^abc^	19.93 ± 4.03 ^abc^
	23.50–26.00	13.25–21.82	13.10–18.03	8.81–11.36	5.00–7.82	2.77–3.96	2.45–3.27	1.00–1.40	27.63–50.42	13.11–27.82
	3.4	15.00%	9.90%	6.90%	13.20%	10.50%	9.30%	9.50%	18.00%	20.20%
*R. sativus* subsp. *sinensis* Sazon. et Stankev. convar. *sinensis*										
var. *roseus* Sazon.										
Red Giant (10)	34.44 ± 1.57 ^i^	21.44 ± 1.86 ^acgi^	18.97 ± 1.48 ^ce^	12.88 ± 1.70 ^hi^	8.16 ± 0.53 ^c^	7.22 ± 2.07 ^e^	2.17 ± 0.24 ^b^	3.35 ± 1.01 ^e^	50.80 ± 18.16 ^de^	27.31 ± 8.79 ^ced^
	32.00–37.50	19.60–25.60	16.70–21.25	10.90–15.20	7.60–9.00	4.20–11.50	1.80–2.44	2.10–4.80	24.00–94.00	12.50–45.63
	4.50%	8.70%	7.80%	13.20%	6.50%	28.70%	11.10%	30.10%	35.80%	32.2%
var. *sinensis*										
Burpee’s white (3)	34.77 ± 0.51 ^i^	21.43 ± 3.11 ^acgij^	17.60 ± 1.20 ^acdeg^	12.33 ± 1.02 ^aceghi^	7.60 ± 0.52 ^aceg^	3.30 ± 0.35 ^a^	3.07 ± 0.32 ^a^	1.07 ± 0.06 ^a^	52.17 ± 5.75 ^de^	28.83 ± 3.86 ^ced^
	34.20–35.20	18.20–24.40	16.40–18.80	11.60–13.50	7.30–8.20	3.10–3.70	2.70–3.30	1.00–1.10	46.40–57.90	25.10–32.80
	1.50%	14.50%	6.80%	8.30%	6.80%	10.50%	10.50%	5.40%	11.00%	13.40%
Mean of collection (149)	24.75 ± 3.63	19.51 ± 3.85	16.46 ± 2.39	10.90 ± 1.78	6.93 ± 1.24	4.48 ± 2.09	2.75 ± 0.55	1.81 ± 1.30	37.75 ± 12.67	20.86 ± 7.33
	20.50–37.50	10.17–30.50	11.58–21.72	6.33–15.20	4.33–9.90	2.34–13.83	1.41–5.36	0.81–9.82	17.69–114.50	10.22–75.20
	14.70%	19.70%	14.50%	16.30%	17.90%	46.70%	19.80%	71.60%	33.60%	35.20%
*R. sativus* subsp. *sativus* convar. *sativus*										
var. *sativus*										
Odesskaya (4)	34.88 ± 1.45 ^f^	28.78 ± 4.93 ^abd^	31.00 ± 3.36 ^acde^	14.30 ± 1.18 ^ab^	8.60 ± 0.52 ^de^	6.80 ± 0.74 ^ef^	6.65 ± 0.66 ^abcdef^	1.05 ± 0.13 ^be^	385.08 ± 90.74 ^ab^	230.22 ± 70.32 ^abcd^
	33.80–37.00	21.60–32.70	27.10–35.00	12.60–15.20	8.20–9.30	6.30–7.90	5.80–7.30	0.90–1.20	259.10–465.00	146.80–312.00
	4.10%	17.10%	10.80%	8.30%	6.10%	10.90%	9.90%	12.30%	23.60%	30.60%
var. *rubrus* Sinsk.										
Ostergrus (8)	35.06 ± 0.73 ^f^	18.25 ± 2.86 ^be^	24.80 ± 2.86 ^af^	13.69 ± 1.77 ^ab^	8.46 ± 1.17 ^de^	14.25 ± 7.91 ^abcdg^	3.96 ± 0.96 ^cg^	4.07 ± 2.96 ^cd^	231.75 ± 108.32 ^b^	131.90 ± 70.54 ^abc^
	34.00–36.50	14.29–22.50	21.00–29.37	11.25–16.92	6.75–10.42	6.14–31.38	2.70–4.90	1.32 ± 10.24	113.40–396.20	69.00–278.40
	2.10%	15.70%	11.50%	13.00%	13.90%	55.50%	24.20%	52.90%	46.70%	53.50%
convar. *hybernus* (Alef.) Sazon. var *niger* (L.) Sinsk.										
Round Black Spanish (14)	79.32 ± 9.13 ^e^	29.34 ± 9.38 ^abd^	33.72 ± 6.15 ^acd^	16.56 ± 3.16 ^ab^	11.20 ± 1.61 ^ab^	6.87 ± 1.12 ^e^	7.23 ± 0.91 ^d^	0.94 ± 0.09 ^e^	401.88 ± 153.38 ^ab^	258.71 ± 101.52 ^abc^
	61.00–87.00	22.70–57.75	27.80–45.04	14.00–25.71	10.20–16.46	5.30–9.10	6.00–8.95	0.80–1.10	245.10–678.56	153.20–454.30
	11.50%	32.00%	18.20%	19.10%	14.40%	16.40%	12.60%	9.10%	38.20%	39.20%
Long Black Spanish (2)	65.00 ± 0.00 ^d^	36.53 ± 3.25 ^acd^	42.82 ± 3.98 ^bd^	21.06 ± 3.74 ^acd^	10.19 ± 1.29 ^abcde^	24.48 ± 6.06 ^cd^	4.77 ± 0.23 ^abfg^	5.17 ± 1.52 ^cd^	573.90 ± 61.72 ^ab^	387.08 ± 43.50 ^abcd^
	65.00–65.00	34.23–38.83	40.00–45.63	18.42–23.70	9.27–11.10	20.20–28.76	4.61–4.93	4.10–6.24	530.26–617.54	356.32–417.84
	0.00%	8.90%	9.30%	3.70%	12.70%	24.80%	4.80%	29.40%	10.80%	11.20%
var. *hybernus*										
Winter white round (5)	77.40 ± 14.02 ^e^	29.15 ± 9.41 ^abd^	34.78 ± 3.65 ^acd^	15.58 ± 3.93 ^ab^	11.25 ± 1.81 ^abc^	7.88 ± 0.66 ^bef^	7.66 ± 0.22 ^de^	1.02 ± 0.11 ^be^	475.48 ± 51.53 ^ab^	309.73 ± 31.06 ^abcd^
	52.50–86.00	21.80–44.54	32.30–41.00	12.00–22.29	10.00–14.33	6.90–8.40	7.40–8.00	0.90–1.10	424.90–556.00	268.30–355.80
	18.10%	32.30%	10.50%	25.20%	16.10%	8.30%	2.90%	10.50%	10.80%	10.00%
var. *violaceus*										
Violet d’hiver de Gournay (3)	65.00 ± 0.00 ^d^	42.88 ± 6.13 ^c^	47.43 ± 2.39 ^b^	21.62 ± 2.06 ^c^	11.71 ± 1.35 ^a^	20.66 ± 4.47 ^abcd^	5.30 ± 0.54 ^abcg^	3.91 ± 0.89 ^abcd^	574.38 ± 201.11 ^ab^	361.52 ± 151.95 ^abcd^
	65.00–65.00	36.65–48.90	44.74–49.33	19.25–22.90	10.55–13.19	15.50–23.39	4.86–5.91	3.02–4.81	379.88–781.49	211.88–515.67
	0.00%	14.30%	5.00%	9.50%	11.50%	21.60%	10.30%	22.80%	35.00%	42.00%
*R. sativus* subsp. *sinensis* convar. *lobo* Sazon. et Stankev.										
var. *lobo*										
Chinto Lobe (17)	47.90 ± 7.26 ^a^	25.97 ± 6.97 ^abd^	30.03 ± 4.54 ^ac^	15.25 ± 2.63 ^ab^	10.35 ± 1.68 ^ade^	14.11 ± 3.02 ^a^	5.83 ± 1.23 ^a^	2.63 ± 1.06 ^a^	476.43 ± 113.42 ^a^	281.14 ± 82.07 ^abcd^
	37.50–63.50	18.90–49.27	21.20–41.18	12.40–23.52	8.50–15.28	8.00–19.40	3.50–8.85	1.17–5.50	362.40–691.50	124.40–418.27
	15.20%	26.80%	15.10%	17.30%	16.20%	21.40%	21.00%	40.30%	23.80%	29.20%
var. *virens* Sazon.										
Margilanskaya (8)	47.56 ± 3.06 ^abc^	25.62 ± 3.67 ^abd^	30.27 ± 3.20 ^acde^	15.19 ± 2.96 ^ab^	10.55 ± 0.93 ^abc^	11.07 ± 2.70 ^abcfg^	5.73 ± 0.42 ^abcf^	1.98 ± 0.64 ^abc^	399.12 ± 93.44 ^ab^	231.68 ± 38.98 ^abc^
	41.70–51.00	21.20–33.39	26.30–36.24	11.50–20.87	9.60–12.20	8.00–16.20	5.00–6.40	1.30–3.30	275.10–571.70	161.10–299.00
	6.40%	14.30%	10.60%	19.50%	8.90%	24.40%	7.30%	32.50%	23.40%	16.80%
var. *rubidus* Sazon.										
Ashkhabadskaya (4)	52.63 ± 2.95 ^ab^	22.80 ± 1.09 ^abde^	28.43 ± 1.39 ^acd^	13.35 ± 0.58 ^ab^	9.55 ± 0.40 ^abcde^	13.40 ± 0.81 ^ab^	5.93 ± 0.43 ^ab^	2.28 ± 0.22 ^ab^	377.75 ± 64.63 ^ab^	278.50 ± 53.55 ^abcd^
	48.50–55.00	21.30–23.90	27.00–29.90	12.80–14.10	9.20–10.10	12.40–14.20	5.40–6.40	2.00–2.50	282.70–420.40	199.60–318.70
	5.60%	4.80%	4.9%	4.40%	4.20%	6.10%	7.20%	9.80%	17.10%	19.20%
Hun-dong-lon (9)	47.36 ± 4.73 ^ac^	32.10 ± 10.92 ^ab^	36.20 ± 10.29 ^acde^	17.86 ± 5.85 ^abd^	11.71 ± 2.77 ^ab^	7.64 ± 1.02 ^efg^	6.94 ± 1.05 ^abcdf^	1.12 ± 0.26 ^be^	438.64 ± 160.79 ^ab^	251.74 ± 90.08 ^abc^
	41.70–57.50	18.10–49.83	26.00–57.30	9.30–25.00	8.50–15.60	6.10–9.40	5.30–8.28	0.90–1.70	224.60–616.40	116.60–351.22
	10.00%	34.00%	28.40%	32.80%	23.70%	13.40%	15.10%	22.90%	36.70%	35.80%
Shun-hun-pan-chun-zy (4)	45.08 ± 6.06 ^abc^	29.35 ± 5.82 ^abd^	31.15 ± 4.05 ^acde^	20.18 ± 4.72 ^cd^	10.78 ± 1.22 ^abcde^	14.40 ± 1.94 ^abcdfg^	5.45 ± 0.87 ^abcfg^	2.69 ± 0.54 ^abcd^	460.37 ± 207.86 ^ab^	321.30 ± 175.26 ^abcd^
	37.50–51.50	23.10–35.52	27.10–36.56	16.80–27.17	9.40–11.85	11.90–16.50	4.26–6.25	2.00–3.31	298.60–765.25	210.36–582.93
	13.50%	19.80%	13.00%	23.40%	11.30%	13.40%	15.90%	20.10%	45.20%	54.60%
var. *incarnatus* Sazon.										
Xing-li-mei (7)	45.57 ± 8.98 ^abc^	24.40 ± 6.32 ^abde^	37.08 ± 7.57 ^cde^	19.13 ± 3.54 ^cd^	10.49 ± 2.22 ^abcd^	7.90 ± 2.10 ^bcefg^	6.27 ± 1.01 ^abcdef^	1.25 ± 0.17 ^be^	447.21 ± 84.41 ^ab^	230.34 ± 117.95 ^abd^
	37.00–60.00	16.00–35.25	25.50–48.63	14.00–25.29	7.50–14.50	5.67–12.00	5.00–8.00	1.03–1.50	348.67–570.00	125.00–475.00 c
	19.70%	25.90%	20.40%	18.50%	21.20%	26.70%	16.20%	13.80%	18.90%	51.20%
var. *syringeus* Sazon.										
Purple (3)	48.00 ± 17.09 ^abc^	23.72 ± 3.87 ^abde^	27.51 ± 4.90 ^acd^	16.90 ± 1.13 ^abc^	9.51 ± 0.76 ^ade^	16.10 ± 7.71 ^abc^	5.03 ± 1.80 ^abcg^	3.75 ± 2.35 ^abc^	352.65 ± 69.64 ^ab^	198.60 ± 77.19 ^abcd^
	30.00–64.00	21.00–28.15	23.67–33.03	16.17–18.21	8.83–10.33	7.40–22.10	3.80–7.10	1.04–5.26	309.60–433.00	109.80–249.60
	35.60%	16.30%	17.80%	6.70%	8.00%	47.90%	35.80%	42.70%	19.80%	38.90%
*R. sativus* subsp. *acanthiformis* (Blanch.) Stankev. convar. *minowase* (Kitam.) Sazon.										
var. *minowase*										
Kameido (8)	45.33 ± 3.83 ^ac^	24.94 ± 3.86 ^abd^	29.68 ± 4.82 ^acdef^	15.26 ± 3.05 ^ab^	9.69 ± 1.44 ^acde^	17.09 ± 3.91 ^bcd^	5.75 ± 1.15 ^abcf^	3.17 ± 1.22 ^abcd^	499.26 ± 129.07 ^a^	323.30 ± 102.27 ^acd^
	39.00–51.00	18.40–31.50	24.30–39.61	12.30–20.47	7.90–12.73	11.42–22.90	3.90–8.02	1.42–5.50	354.70–722.95	176.90–501.86
	8.40%	15.50%	16.25%	20.00%	14.80%	22.90%	20.00%	38.70%	25.90%	31.60%
Minovase (13)	44.54 ± 5.34 ^ac^	27.37 ± 4.10 ^abd^	32.15 ± 5.77 ^acde^	17.38 ± 4.30 ^abcd^	10.76 ± 1.99 ^abc^	22.82 ± 7.81 ^cd^	5.13 ± 0.76 ^bcg^	4.56 ± 1.39 ^cd^	616.90 ± 334.28 ^a^	439.54 ± 279.39 ^d^
	36.25–56.00	20.60–36.75	24.30–43.88	10.00–28.00	8.30–14.76	13.30–37.64	3.90–6.20	2.10–6.32	324.00–1370.50	156.90–1057.13
	12.00%	15.00%	17.90%	24.80%	18.50%	34.20%	14.80%	30.50%	54.20%	63.60%
Ninengo (4)	47.30 ± 1.41 ^abc^	25.03 ± 7.03 ^abd^	27.83 ± 5.82 ^acdef^	14.65 ± 3.17 ^abcd^	9.63 ± 1.21 ^abcde^	23.28 ± 2.39 ^cd^	4.45 ± 0.98 ^bcg^	5.58 ± 1.73 ^cd^	501.90 ± 82.57 ^a^	264.48 ± 133.86 ^abcd^
	46.00–49.00	19.00–34.30	22.30–35.80	11.50–18.50	8.10–10.90	21.00–26.40	3.00–5.20	4.10–8.00	422.70–615.30	80.50–400.70
	3.00%	28.10%	20.90%	21.60%	12.60%	10.30%	22.10%	31.10%	16.50%	50.60%
convar. *acanthiformis*										
Miyashige (11)	45.84 ± 5.21 ^ac^	24.47 ± 3.84 ^abd^	27.12 ± 2.86 ^aef^	14.58 ± 2.94 ^ab^	9.68 ± 1.25 ^acde^	18.72 ± 5.03 ^bcd^	5.12 ± 0.76 ^abc^	3.79 ± 1.22 ^cd^	444.96 ± 169.67 ^ab^	297.25 ± 116.94 ^acd^
	39.00–57.00	19.30–30.90	22.00–31.40	11.90–20.75	7.80–12.1	10.70–26.93	3.80–6.60	1.80–5.30	316.00–892.90	167.90–587.70
	11.40%	15.70%	10.50%	20.20%	12.90%	26.90%	14.80%	32.10%	38.10%	39.30%
Shiroagiri (5)	41.25 ± 4.19 ^c^	25.62 ± 2.76 ^abd^	30.15 ± 0.93 ^acdef^	15.76 ± 3.23 ^ab^	9.24 ± 0.91 ^de^	14.86 ± 4.35 ^abcd^	5.75 ± 0.53 ^abcf^	2.61 ± 0.97 ^abcd^	453.38 ± 83.97 ^a^	323.39 ± 93.33 ^ad^
	35.25–44.90	21.90–29.20	29.40–31.50	13.70–21.49	8.20–10.61	8.30–19.32	4.87–6.30	1.40–3.97	313.20–512.60	170.50–421.90
	10.20%	10.80%	3.10%	20.50%	9.90%	29.30%	9.20%	37.10%	18.50%	28.90%
Mean of collection (129)	50.84 ± 14.28	26.0 ± 7.27	31.56 ± 6.53	16.20 ± 3.70	10.30 ± 1.76	14.00 ± 6.81	5.82 ± 1.27	2.71 ± 1.76	451.42 ± 175.82	284.93 ± 140.19
	30.00–87.00	14.29–57.75	21.00–57.30	9.30–28.00	6.75–16.46	5.30–37.64	2.70–8.95	0.80–10.24	113.40–1370.50	69.00–1057.19
	28.10%	27.10%	20.70%	22.80%	17.10%	48.60%	21.90%	65.20%	39.00%	49.20%

Note: All data are presented as Mean ± SD, Xmin-Xmax, CV, %; ^a–j^ Values with different superscript in the column were significantly different (*p* < 0.05).

**Table 2 plants-10-01799-t002:** Characteristics of subspecies of small radish and radish by main biochemical parameters.

Trait	Small Radish			Radish			
	subsp. *sativus*	subsp. *sinensis*	Mean	subsp. *sativus*	subsp. *sinensis*	subsp. *acanthiformis*	Mean
Dry matter. %	5.59 ± 1.19 ^a^	7.04 ± 0.57 ^b^	5.67 ± 1.21 *	10.44 ± 1.63 ^a^	7.90 ± 1.81 ^b^	7.06 ± 0.99 ^c^	8.14 ± 1.96 **
	3.72–8.56	6.52–8.24	3.72–8.56	7.50–13.52	5.28–13.64	5.56–10.52	5.28–13.64
	21.34%	8.09%	21.36%	15.57%	22.89%	14.00	24.10%
Ascorbic acid. mg/100 g	30.58 ± 6.20 ^a^	29.41 ± 8.30 ^a^	30.47 ± 6.29 *	37.78 ± 14.58 ^a^	41.84 ± 16.51 ^a^	44.71 ± 14.82 ^a^	42.02 ± 15.64 **
	19.04–57.72	24.48–48.10	19.04–57.72	14.00–56.40	15.00–73.32	11.80–62.04	11.80–73.32
	20.27%	28.24%	20.63%	38.59%	39.69%	33.14%	37.22%
Monosaccharides. %	1.83 ± 0.99 ^a^	2.97 ± 1.25 ^b^	1.89 ± 1.04 *	2.48 ± 0.50 ^a^	1.95 ± 0.^67 ab^	1.80 ± 0.58 ^b^	2.00 ± 0.65 *
	0.07–4.35	1.38–4.64	0.07–4.64	1.50–3.02	0.85–3.18	1.00–2.61	0.85–3.18
	54.11%	42.25%	54.89%	20.30%	34.39%	32.09%	32.37%
Total sugar content. %	2.06 ± 1.06 ^a^	3.45 ± 1.54 ^b^	2.14 ± 1.14 *	2.93 ± 0.38 ^a^	2.27 ± 0.82 ^ab^	2.11 ± 0.78 ^b^	2.34 ± 0.79 *
	0.16–5.00	1.60–5.45	0.16–5.45	2.47–3.59	1.10–4.31	1.03–3.39	1.03–4.31
	51.46%	44.66%	53.20%	13.08%	36.03%	36.84%	33.63%

Note: All data are presented as Mean ± SD, X_min_–X_max_, CV, %; ^a–c^; *-** Values with different superscript in the column were significantly different (*p* < 0.05).

**Table 3 plants-10-01799-t003:** Factor structure of variability of features of 128 accessions of *R. sativus.*

Variable	Factor Loadings (Varimax Raw) Extraction: Principal Components			
	PC 1	PC 2	PC 3	PC 4
Rosette height	0.84	0.10	−0.19	0.20
Rosette diameter	0.85	0.19	0.03	0.40
Leaf length	0.88	0.20	−0.20	0.24
Leaf width	0.86	0.17	−0.01	0.32
Root length	0.49	0.83	0.19	0.06
Root diameter	0.80	−0.13	0.35	0.40
Root index	0.05	0.98	−0.05	0.01
Plant weight	0.83	0.40	0.19	0.21
Root weight	0.83	0.38	0.20	0.15
Duration of vegetative phase	0.54	0.07	0.32	0.70
Dry matter	0.31	0.03	0.05	0.93
Ascorbic acid	−0.02	0.08	0.94	0.13
Expl.Var	5.61	2.10	1.31	1.96
Prp.Totl	0.47	0.17	0.11	0.16
% total variance	61.32	14.14	10.88	5.17
% cumulative proportion of variance	61.32	75.46	86.34	91.51

## Data Availability

The data presented in this study are available in article.

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
