# Peer review of "Genetic Diversity of Phenotypic and Biochemical Traits in VIR Radish (Raphanus sativus L.) Germplasm Collection"

_plants, 2021, doi:10.3390/plants10091799_

Round 1

Reviewer 1 Report

The insertion of graphs into the manuscript as recommended during the first review has improved the quality of the paper.

Although the revised manuscript shows some improvement in the use of the English language and style, further editing of the entire text is required. Specific comments and recommendations made during the first review, have not been addressed one by one in the reply to the reviewer and only some of those seem to have been addressed in the revised version.

L281-289 (previously L199-200): Among the studied accessions, promising for breeding use are accessions of the Saxa type from the Netherlands (KD, k-2167;

On what basis did you decide that these accessions are promising for breeding and for which major traits? This should be explained in the text.

This is a concern from the first review which has not yet been addressed.

L357-360 (previously L267-270): As a result of the small radish and radish harvest component studying, the amplitude of variation in the plant weight (17.69-114.50 g and 113.40-1370.50 g) and a root weight (10.22-75.20 and 69.00-1057, 19 d), which had a high degree of variability (Cv more than 30%).

Sentence not complete. Please check.

This is a concern from the first review which has not yet been addressed.

L484-485: In a small radish, 7 groups with a step of 0.76% have been identified according to the sugar content.

As the small radish group is meant here, the ‘a’ in the first part of the sentence should be deleted to read: In small radish, 7 groups…

This is a concern from the first review which has not yet been addressed.

L548-549: the diameter of the root crop and the weight of the plant and the root

I believe you mean the diameter of the root, not of the ‘root crop’.

L596: The role of world resources for economic use has been repeatedly emphasized by…

What do you mean by world resources for economic use? Are you referring to commodities or to genetic resources?

L619: and thus may not provide sufficient understanding of the diversification of R. sativus

‘diversification’ should be replaced by ‘genetic diversity’ of…

L637-638: The Department of Vegetable crops and Cucurbits of the VIR is working on the creation of traits collections based on the extensive collection of R. sativus of the Institute [52, 639 53].

Apart from trait collections, core collections would by highly useful to facilitate germplasm use for breeding and research. This aspect is still missing in the revised discussion section. See, for example, Lee, Y. J., Mun, J. H., Jeong, Y. M., Joo, S. H., & Yu, H. J. (2018). Assembly of a radish core collection for evaluation and preservation of genetic diversity. Horticulture, Environment, and Biotechnology59(5), 711-721.

The benefit of using molecular markers to assess genetic diversity of crops has been mentioned in the Discussion section, and it is obvious that a combination of morphological, biochemical and molecular markers would be beneficial for the assessment of the genetic diversity of any crop collection. However, it is not clear why VIR is not considering such a combined approach for maximizing the information on the so-called world Raphanus collection maintained by VIR.

L648-650: It is necessary for the creation and use of trait (phenotypic) collections, which is the most effective way one of the possible approaches to structure collections of plant genetic resources of the any crop.

As mentioned above, apart from trait collections, core collections are truly useful for breeding purposes and have been established for large collections of a number of crops. This aspect still needs to be discussed in this section.

L713-714: The selection of the number of optimal factors is carried out using the Scree test.

Reference missing.

L812-813: 37. Victor, K.; Ryabchoun, R.L. Boguslavskiy Wheat and triticale genetic resources in Ukraine Cereal genetic resources in Europe, in Cereal Genetic Resources in Europe.

There seems to be something wrong with the title of this publication.

Author Response

Answers to reviewers on the manuscript " Genetic diversity of radish (Raphanus sativus L.) VIR germplasm collection by phenotypic and biochemical traits"

The authors thank the reviewer for the thorough evaluation of our manuscript and the valuable comments and suggestions. Please find below the point-by-point explanations of the revisions in the manuscript.

The insertion of graphs into the manuscript as recommended during the first review has improved the quality of the paper.

Although the revised manuscript shows some improvement in the use of the English language and style, further editing of the entire text is required. Specific comments and recommendations made during the first review, have not been addressed one by one in the reply to the reviewer and only some of those seem to have been addressed in the revised version.

L281-289 (previously L199-200): Among the studied accessions, promising for breeding use are accessions of the Saxa type from the Netherlands (KD, k-2167;

On what basis did you decide that these accessions are promising for breeding and for which major traits? This should be explained in the text.

This is a concern from the first review which has not yet been addressed.

 We added an explanation to the text: As a result of our study, we found accessions of small radish with a stable manifestation of the desired traits of the leaf apparatus, which can be used as a starting material for breeding.

L357-360 (previously L267-270): As a result of the small radish and radish harvest component studying, the amplitude of variation in the plant weight (17.69-114.50 g and 113.40-1370.50 g) and a root weight (10.22-75.20 and 69.00-1057, 19 d), which had a high degree of variability (Cv more than 30%).

Sentence not complete. Please check.

This is a concern from the first review which has not yet been addressed.

 We wrote a proposal differently: The variability of plant weight in small radish and radish (17.69-114.50 g and 113.40-1370.50 g) and the root weight (10.22-75.20 g and 69.00-1057.19 g) was high (CV over 30%).

L484-485: In a small radish, 7 groups with a step of 0.76% have been identified according to the sugar content.

As the small radish group is meant here, the ‘a’ in the first part of the sentence should be deleted to read: In small radish, 7 groups…

This is a concern from the first review which has not yet been addressed.

 We corrected it. 

L548-549: the diameter of the root crop and the weight of the plant and the root

I believe you mean the diameter of the root, not of the ‘root crop’.

 We corrected it. 

L596: The role of world resources for economic use has been repeatedly emphasized by…

What do you mean by world resources for economic use? Are you referring to commodities or to genetic resources?

We have corrected to ‘world genetic resources’

L619: and thus may not provide sufficient understanding of the diversification of R. sativus

‘diversification’ should be replaced by ‘genetic diversity’ of…

We corrected it. 

L637-638: The Department of Vegetable crops and Cucurbits of the VIR is working on the creation of traits collections based on the extensive collection of R. sativus of the Institute [52, 639 53].

Apart from trait collections, core collections would by highly useful to facilitate germplasm use for breeding and research. This aspect is still missing in the revised discussion section. See, for example, Lee, Y. J., Mun, J. H., Jeong, Y. M., Joo, S. H., & Yu, H. J. (2018). Assembly of a radish core collection for evaluation and preservation of genetic diversity. Horticulture, Environment, and Biotechnology59(5), 711-721.

The benefit of using molecular markers to assess genetic diversity of crops has been mentioned in the Discussion section, and it is obvious that a combination of morphological, biochemical and molecular markers would be beneficial for the assessment of the genetic diversity of any crop collection. However, it is not clear why VIR is not considering such a combined approach for maximizing the information on the so-called world Raphanus collection maintained by VIR.

Thank you for your comment. Really creating core collections one of the main tasks of studying and structuring the VIR collections. The first step of creating trait and core collections is the study of phenotypic traits, as well as physiological and biochemical traits. The next step in advanced learning is genotyping. We've added information about the core collections to the discussion section. VIR is working with molecular markers to create the core collections. We have added this information to the discussion section.

L648-650: It is necessary for the creation and use of trait (phenotypic) collections, which is the most effective way one of the possible approaches to structure collections of plant genetic resources of the any crop.

As mentioned above, apart from trait collections, core collections are truly useful for breeding purposes and have been established for large collections of a number of crops. This aspect still needs to be discussed in this section.

We have added this information to the discussion section.

L713-714: The selection of the number of optimal factors is carried out using the Scree test.

Reference missing.

We have added reference

L812-813: 37. Victor, K.; Ryabchoun, R.L. Boguslavskiy Wheat and triticale genetic resources in Ukraine Cereal genetic resources in Europe, in Cereal Genetic Resources in Europe.

There seems to be something wrong with the title of this publication.

We corrected it. 

Reviewer 2 Report

The manuscript has been significantly improved and is recommended for publication. Added some comments for authors to review and use.

Author Response

The authors thank the reviewer for the thorough evaluation of our manuscript and the valuable comments and suggestions. We have made the necessary changes to the text in accordance with your edits

This manuscript is a resubmission of an earlier submission. The following is a list of the peer review reports and author responses from that submission.

Round 1

Reviewer 1 Report

Manuscript No. plants-1291813

Title: Genetic diversity and biochemical value of radish (Raphanus sativus L.) VIR germplasm collection

Comment:

In this study, authors analyzed the evaluation of phenotype and biochemicals in 149 radish accessions collected form 37 countries. It is important to understand the morphological traits and chemical composition of plant germplasms to plan further breeding program.

However, authors simply explained their results and did not fully discuss of the results.

In addition, it needs sufficient explanation of the radish accessions used in this study (eg. Why 149 accessions from 37 countries were selected in this study), for readers.

Reviewer 2 Report

General observations

While this study describes in detail variations in phenological, morphological and nutritional traits, there is an obvious lack of graphical illustrations to facilitate reading and capturing the salient findings of the paper. It is suggested to add some graphs that highlight, for example, variation of major traits of interest for breeding in the two crops, radish and small radish, capturing those accessions which are highlighted as having great breeding value. Currently, the text is very tiring to read, and it is difficult to capture the salient points.

The VIR Raphanus collection consists of 1600 small radish and 1200 radish accessions as mentioned in the Abstract. It is important to describe in the Materials and Methods section the criteria used to select the sub-populations of 149 small radish accessions and 129 radish accessions for this study.

From line 199 onwards, you mention several accessions which are promising for breeding. However, it is not clear on what basis it was decided that these accessions are promising for breeding and for which major traits. This should be explained in the text.

You mention the creation and use of trait collections in this paper. Do you mean to create core and/or mini-core collections within the VIR Raphanus collection based on traits of interest for breeding to facilitate germplasm use? Which types of traits do you have in mind? This concept of trait collections should be properly introduced, and examples given to be more meaningful. How would the creation of trait collections help preserve the genetic diversity of the Raphanus gene pool as mentioned in the Abstract? This statement made in the Abstract needs some supportive arguments/explanations in the text!

In its present form, the Discussion section is not acceptable. It is a noticeably short section that briefly mentions other studies on the influence of abiotic factors on the growth and development of Raphanus, the creation of CMS lines, and molecular marker and genomic studies. These are all topics that are not dealt with in this paper, hence do not merit to be mentioned in the Discussion section. They could be briefly mentioned in the Introduction section but are not relevant for this paper. The focus of this paper, the variations observed in phenological, morphological and nutritional traits within the VIR Raphanus collection and their importance for breeding are not discussed at all in this section. This needs to be corrected before the paper is publishable in the journal Plants.

The entire text requires careful editing of the English language and style. Some details are highlighted below.

Specific comments/suggestions:

L9: in VIR genebanc,

Please correct to ‘genebank’

L15-16: Small radish set was included 149 accessions from 37 countries

Suggested edit: The small radish subset included…..

L17: radish set was included 129 accessions from 21 countries,

Suggested edit: The radish subset included ….

L19-20: phonological, morphological and biochemical characteristics,

Phonological characteristics = sound characteristics of a language!

Do you perhaps mean: ‘phenological’?

L24-26: It is necessary for the creation and use of trait collections, which is the most effective way to structure collections and preserve the genetic diversity of the gene pool

Suggested edit: Such information is of importance for the creation and use of trait collections. Trait collections facilitate germplasm use and help preserve the genetic diversity of the gene pool.

L28: based on the 27 results of the investigation.

Do you mean: ‘based on the results of this investigation?

L107-108: European radish has a large distribution and provides a high yield in almost regions of Russian Federation.

Do you mean: ….”in almost all regions of the Russian Federation’?

L111-112: s during the winter and spring times [35, 36]

Suggested edit: replace ‘times’ with ‘months’.

L125: valuable for breeding, ecological plasticity,

I believe you mean: ‘value for breeding’ or ‘breeding value’

L130: The world collection of Raphanus L. root crops, maintained in VIR genebank

As you designate the VIR Raphanus collections as ‘world collection’, it would be interesting for the reader to mention a few other important Raphanus collections, such as those in NIAS Japan, IPK Germany, NE9 USA and NBPGR India.

L141-142: and to identify the degree of variability of traits due to botanical and systematical status of samples

This part of the sentence is not clear. Please clarify!

L146-147: The duration of the vegetative period is one of the most important valuable traits for.

For what?

L151-152: The small radish collection was divided into 11 groups with a step of 1.55

What kind of steps are these – days?

L155, heading of Table 1 column: Period of vegetative, days

Suggested edit: Duration of vegetative phase, days

L169: significantly difference was found between

Please correct ‘significantly’ to ‘significant’

L199-200: Among the studied accessions, promising for breeding use are accessions of the Saxa type from the Netherlands (KD, k-2167;

On what basis did you decide that these accessions are promising for breeding and for which major traits? This should be explained in the text.

L255: a wide variety was also noted in length (5.30-

I believe you mean root length here. Please insert ‘root’ before length.

L259: and the by the root - into 8 groups with a step of 1.18.

It is not clear what is meant with ‘the by the root’ – please clarify.

L267-270: As a result of the small radish and radish harvest component studying, the amplitude of variation in the plant weight (17.69-114.50 g and 113.40-1370.50 g) and a root weight (10.22-75.20 and 69.00-1057, 19 d), which had a high degree of variability (Cv more than 30%).

Sentence not complete. Please check.

L272: in small radish was reviled in types with a round

Do you mean ‘revealed’ instead of ‘reviled’?

L304: , is significantly differ in all the studied traits from

Suggested edit: Either: ‘is significantly different’… or ‘differs significantly…’

L351-352: Depending on the variety, the sugar content in the dry matter can reach 25-55%. In our studies, the sugar content in the dry matter was 3.5-76.9% in small radish.

The second statement (3,5-76,9% dry matter) seems to contradict the first statement that dry matter can reach up to 55%.

L357: In a small radish 7 groups with a step of 0.76 have been identified according to the sugar content.

Suggested edit: In small radish, 7 groups….

L403-408: This is a very long sentence and it is suggested to break it up into two to facilitate reading.

L424-428: Our study of the degree of variation of phenotypic indicators allowed us to identify adaptive stable and highly variable traits and properties of R. sativus accessions, which is necessary for the creation and use of the trait collections, which are the very effective way to structure collections, preserve the genetic diversity of the gene pool and expand the range of variability of the source material for the breeding.

You mention the creation and use of trait collections. Do you mean to create core and/or mini-core collections within the VIR Raphanus collection based on traits of interest for breeding to facilitate germplasm use? Which types of traits do you have in mind? This concept of trait collections should be properly introduced, and examples given to be more meaningful. How would the creation of trait collections help preserve the genetic diversity of the Raphanus gene pool as mentioned in the Abstract? This statement made in the Abstract needs some supportive arguments/explanations in the text!

L435-439: The studied small radish sub-collection was represented by 149 accessions from 37 countries, belonging to 13 types of 7 varieties of European and Chinese subspecies; the radish sub-collection was represented by 129 accessions from 21 countries, belonging to 18 types of 11 varieties of European, Chinese and Japanese subspecies.

The VIR Raphanus collection consists of 1600 small radish and 1200 radish accessions as mentioned in the Abstract. It is important to mention in the Materials and Methods section the criteria used to select the sub-populations of 149 small radish accessions and 129 radish accessions for this study.

L451-452: Sowing in the field was carried out in mid-July, after the end of the period of white nights.

What is the meaning of ‘white nights’? This requires a clarification!

L479-481: sugars – by the Bertrand method; ascorbic acid – by direct extraction from plants with 1% hydrochloric acid, followed by titration with 2,6-dichlorindofinol (Tilman’s reagent).

References are required for the methods used.

Reviewer 3 Report

The paper titled "Genetic diversity and biochemical value of radish (Raphanus sativus L.) VIR germplasm collection" is finalized to describe the radish collection belonging to the VIR repository.

The paper, from the title to the text, is referred to genetic diversity but without any molecular information and with only morphological traits and very few chemical data. The paper need several improvements in all chapters. The manuscript is (more or less) only a list of phenotypic traits (in the long 2.1 chapter) without any correlation analysis (e.g. PCA) or strong conclusion. The chemical traits are referred to only 3 class of compounds and, again, without correlation analysis between them and morphological results.

In addition to the low (new) information belonging to this manuscript, in my opinion (this is mandatory) the authors must to add a molecular characterization with available and common used markers. 

As reported in previous comments, the authors spoke about genetic diversity (also in the very poor discussion…e.g. "to preserve the genetic diversity of the gene pool and expand the range of variability of the source material for the breeding") without any genetic information or data…therefore the paper cannot be accepted.

Reviewer 4 Report

An interesting study and a great collection of genetic resources included in it. The research was carried out in accordance with the set goal, using an appropriate methodology. The obtained results and conclusions will definitely be useful for both germplasm conservation activities and selection.

For a better presentation of the results and for improving the quality of the manuscript, I would have some general suggestions that should be taken into account.

Introduction - The description is very redundant in describing previous studies, the scientists involved. This section could be shortened, by excluding the glorification of specific scientists, leaving only references to their work and mentioning the results. At the same time, the introduction lacks information on radish cultivation and distribution in the world. Global statistics, major growers, etc. should be provided so that the reader understands the importance of this crop.

Results - Throughout the text, cultivar names should be enclosed in single quotation marks, it is currently difficult to understand what is a group of different genotypes and what is an advanced, purposefully bred cultivar. The results of both morphology and chemical composition lack some kind of multidimensional analysis, which would allow assessing the grouping of samples by all traits together. At present, the grouping by individual traits is described in detail, but there is no overview. This would be very useful in the germplasm assessment.

Discussion - The discussion currently describes general statements that do not follow from this particular study (some of which are not even related to it). Therefore, the discussion should be modified so that the results obtained in it (which are very good) are compared with other studies, whether the results of this study confirm what is found in other studies or, conversely, what is new, etc. The discussion should be a transitional section between the results and the conclusions. I think that this section should be substantially revised.

References - The number of references is large for a research publication (70), however, a large part (23 or 32%) is in Russian and is not available to most readers. Perhaps they can be replaced, at least in part, by a smaller number of publications in English (if possible). In any case, the number of references is large enough to compensate for these as well.

General comments - It would be good to review the manuscript by an English speaking editor because in some places the sentence structure is inappropriate and difficult to understand. Some suggestions for corrections are in the attached file.
